# Asymmetric Dinuclear Lanthanide(III) Complexes from the Use of a Ligand Derived from 2-Acetylpyridine and Picolinoylhydrazide: Synthetic, Structural and Magnetic Studies [note 1]

**DOI:** 10.3390/molecules25143153

**Published:** 2020-07-10

**Authors:** Diamantoula Maniaki, Panagiota S. Perlepe, Evangelos Pilichos, Sotirios Christodoulou, Mathieu Rouzières, Pierre Dechambenoit, Rodolphe Clérac, Spyros P. Perlepes

**Affiliations:** 1Department of Chemistry, University of Patras, 26504 Patras, Greece; dia.maniaki@gmail.com (D.M.); pilvag@gmail.com (E.P.); 2CNRS, Univ. Bordeaux, CNRS, Centre de Recherche Paul Pascal, UMR 5031, 33600 Pessac, France; panagiota.perlepe@crpp.cnrs.fr (P.S.P.); mathieu.rouzieres@crpp.cnrs.fr (M.R.); 3CNRS, Univ. Bordeaux, Bordeaux INP, ICMCB, UMR 5026, 33600 Pessac, France; 4ICFO-Institut de Ciencies Fotoniques, The Barcelona Institute of Nanoscience and Nanotechnology, Castelldefels, 08860 Barcelona, Spain; Sotirios.Christodoulou@alumni.icfo.eu (S.C.); 5Foundation for Research and Technology-Hellas (FORTH), Institute of Chemical Engineering Sciences (ICE-HT), Platani, P.O. Box 1414, 26504 Patras, Greece

**Keywords:** asymmetric dinuclear lanthanide(III) complexes, dysprosium(III) and erbium(III) single-molecule magnets, magnetic properties, metal complexes of *N*’-(1-(1-pyridin-2-yl)ethylidene)pyridine-2-carbohydrazide, magnetization relaxation pathways, single-crystal X-ray structures

## Abstract

A family of four Ln(III) complexes has been synthesized with the general formula [Ln_2_(NO_3_)_4_(L)_2_(S)] (Ln = Gd, Tb, Er, and S = H_2_O; **1**, **2** and **4**, respectively/Ln = Dy, S = MeOH, complex **3**), where HL is the flexible ditopic ligand *N*’-(1-(pyridin-2-yl)ethylidene)pyridine-2-carbohydrazide. The structures of isostructural MeOH/H_2_O solvates of these complexes were determined by single-crystal X-ray diffraction. The two Ln^III^ ions are doubly bridged by the deprotonated oxygen atoms of two “head-to-head” 2.21011 (Harris notation) L¯ ligands, forming a central, nearly rhombic {Ln^III^_2_(μ-OR)_2_}^4+^ core. Two bidentate chelating nitrato groups complete a sphenocoronal 10-coordination at one metal ion, while two bidentate chelating nitrato groups and one solvent molecule (H_2_O or MeOH) complete a spherical capped square antiprismatic 9-coordination at the other. The structures are critically compared with those of other, previously reported metal complexes of HL or L¯. The IR spectra of **1**–**4** are discussed in terms of the coordination modes of the organic and inorganic ligands involved. The f-f transitions in the solid-state (diffuse reflectance) spectra of the Tb(III), Dy(III), and Er(III) complexes have been fully assigned in the UV/Vis and near-IR regions. Magnetic susceptibility studies in the 1.85–300 K range reveal the presence of weak, intramolecular Gd^III^∙∙∙Gd^III^ antiferromagnetic exchange interactions in **1** [*J*/*k*_B_ = −0.020(6) K based on the spin Hamiltonian *Ĥ* = −2*J*(*Ŝ*_Gd1_∙ *Ŝ*_Gd2_)] and probably weak antiferromagnetic Ln^III^∙∙∙Ln^III^ exchange interactions in **2**–**4**. Ac susceptibility measurements in zero dc field do not show frequency dependent out-of-phase signals, and this experimental fact is discussed for **3** in terms of the magnetic anisotropy axis for each Dy^III^ center and the oblate electron density of this metal ion. Complexes **3** and **4** are Single-Molecule Magnets (SMMs) and this behavior is optimally observed under external dc fields of 600 and 1000 Oe, respectively. The magnetization relaxation pathways are discussed and a satisfactory fit of the temperature and field dependencies of the relaxation time *τ* was achieved considering a model that employs Raman, direct, and Orbach relaxation mechanisms.

## 1. Introduction

Devices in the future, such as hard disks, will use components made from molecules instead of the traditional silicon-based electronics [1]. Molecules capable of retaining their magnetization in a given direction were discovered in the 1990s with the now famous {Mn^III^_8_Mn^IV^_4_} acetate complex [2]. Such molecules (single-molecule magnets, SMMs) behave as tiny magnets at very low temperatures. The first generation of SMMs was based on transition-metal complexes [3,4]. Trivalent lanthanides, Ln(III)s, came into the scientific scene of the SMM area in 2003 with the mononuclear, double-decker complexes [Ln^III^(pc)_2_]¯ (Ln = Tb, Dy; pc^2−^ = the phthalocyanine dianion) [5], and they are currently protagonists [6,7,8,9,10,11,12,13,14,15,16,17,18,19,20,21,22,23,24,25,26,27,28,29,30,31,32,33,34] in the interdisciplinary field of molecular magnetism. The latest scientific breakthrough was the synthesis of [(η^5^ − cp*)Dy(η^5^ − cp^iPr5^)][B(C_6_F_5_)_4_], where cp* is the pentamethylcyclopentadienyl(−1) group and cp^iPr5^ is the penta-iso-propylcyclopentadienyl(−1) ligand [35]. This complex displays magnetic hysteresis above liquid-nitrogen temperatures, possessing an effective energy barrier for magnetization reversal, Δ_eff_/*k*_Β’_, of 2217 K and blocking temperatures, *T*_B_ (at 100 s, 65 K) that reach temperatures practical for the development of nanomagnet devices. The great success of Ln(III)s to design SMMs is due to their contracted valence 4f orbitals, a feature that affects both structural and magnetic characteristics [36,37,38]. These orbitals engage in weak, mainly electrostatic interactions with ligands’ orbitals and are thus nearly degenerate in energy, leading to impressive single-ion magnetic anisotropies in Ln(III) complexes. This characteristic is in contrast to d-metal complexes, where covalent interactions between diffuse metal orbitals and ligands quench the orbital angular momentum. Control of the Ln^III^ coordination geometry is thus essential to the construction of SMMs. For example, strongly axial ligand fields can maximize Δ_eff_ for oblate [24] Tb^III^ and Dy^III^ centers and reduce transverse anisotropy, which can in turn decrease the rate of magnetization relaxation through the energy barrier by quantum tunneling. Attaining a high symmetry is also valuable for complexes containing non-Kramers Ln^III^ centers (metal ions with integer spin), e.g., Tb^III^ (4f^8^, *S* = 3) for which ±*M_J_* degeneracy is not guaranteed [38].

Dinuclear Ln(III) complexes are the simplest molecular entities which allow the in-depth investigation of magnetic Ln^III^∙∙∙Ln^III^ exchange interactions [39], which are generally very weak. Scientists have been trying to elucidate the nature and strength of these interactions, as well as the alignment of spin vectors and anisotropy axes; such parameters are affected by the coordination geometries of the Ln^III^ centers, the molecular symmetry and/or the nature of the bridging ligands. Moreover, Ln^III^_2_ SMMs are ideal model systems for providing answers to fundamental questions regarding the paramagnetic slow relaxation from the whole molecule or half of the molecule involving two or a single Ln^III^ center, respectively. Another exciting area to which Ln^III^_2_ complexes are highly relevant is quantum computation [40,41,42]. One of the main strategies is the design of dinuclear complexes of anisotropic metal ions (hence some Ln^III^ ions are ideal candidates), exhibiting dissimilar coordination environments and very weak magnetic coupling. However, the preparation of asymmetric Ln^III^_2_ complexes is by no means easy, because nature often favors symmetric molecules. Asymmetric dinuclear Ln(III) complexes are also valuable, because they allow detailed studies on the two possible thermally activated relaxation processes of molecular origin that are often observed in Ln(III) SMMs [43,44,45].

For the above-mentioned reasons, it is evident that the synthesis of asymmetric Ln^III^_2_ complexes is desirable [42,43,44,45,46]. The design principle behind the implementation of this goal is the appropriate choice of the bridging organic ligand, which should be ditopic. Such a ligand introduces preprogrammed coordination information that is “stored” within its two coordination pockets. When the ligand reacts with a Ln^III^ ion, the reaction interprets this information according to the metal ion’s own coordination “algorithm”. If: (i) the ligand contains one monoanionic bridging atom or group between asymmetrically disposed terminal coordination sites; (ii) it is not very bulky, thus favoring the incorporation of two ligands per dinuclear moiety; (iii) the same pocket of each ligand is coordinated to a single Ln^III^ center (this of course requires a “favorable” serendipity); and (iv) anionic chelating co-ligands are present to saturate the coordination and charge requirements of the Ln^III^ ion, then asymmetric dinuclear complexes will be expected.

The ligand of choice in the present study, which seems to fulfil the first three of the above mentioned criteria, is *N*’-(1-(pyridin-2-yl)ethylidene)pyridine-2-carbohydrazide, shown in its enol-imino form in Figure 1 and abbreviated as HL, and other names of this compound are 2-acetylpyridine picolinoylhydrazone or methyl(pyridin-2-yl)methanone picolinoylhydrazone. This ditopic ligand can be considered as an asymmetric diazine [47] possessing both bidentate and tridentate coordination pockets. The deprotonated ligand (L¯) has two potentially bridging functionalities (μ-*O*, μ-*Ν*-*Ν*) and, due to the free rotation around the single *N-N* bond, can exist in two different coordination conformers, both of which can in principle give dinuclear complexes (but also polynuclear ones). It was hoped that the deprotonated alkoxido oxygen atom (a “hard” base according to the HSAB principle) would be the bridging one towards the Ln^III^ ions (“hard” acids) leaving the closest nitrogen atom of the central backbone free, i.e., non-coordinated; this, indeed, turned out to be the case (vide infra). In addition to the designed criteria discussed above, we decided to work with this ligand for three additional reasons: (a) its published coordination chemistry (a recent report is from our group [48]) remains limited [47,48,49,50,51,52,53,54,55]; (b) three mononuclear Ln(III) complexes, namely [Ln(NO_3_)_3_(HL)(MeOH)_2_] (Ln = La, Ce) and [Nd(NO_3_)_3_(HL)(H_2_O)]∙MeOH, possessing the tridentate chelating N_pyridyl_, N, O (1.10011 using Harris notation [56], Figure 2), neutral ligand in its keto-amino form, had already been structurally characterized [50] (it was our belief that Ln(III) complexes containing the anionic L¯ ligand could exist); and (c) somewhat similar ligands derived from the condensation of picolinic acid hydrazide and other carbonyl-containing compounds have given Ln^III^_2_ complexes with remarkable magnetic properties [43,44].

Taking into account the above information, we report in this paper on the reactions between hydrated gadolinium(III), terbium(III), dysprosium(III) and erbium(III) nitrates and LH, which have led to the desired asymmetric dinuclear complexes. The complexes were characterized by single-crystal X-ray diffraction and conventional spectroscopic methods, while special attention was given to the study of their magnetic properties. This work can be considered as a continuation of the interest of our group in the chemistry, magnetism, and catalytic and photophysical properties of Ln^III^_2_ complexes [57,58,59,60,61,62,63,64,65,66,67], some of which are asymmetric [62,64,65,66].

## 2. Results and Discussion

### 2.1. Synthetic Comments and Conventional Spectroscopic Characterization

We decided to use lanthanide(III) nitrates as starting materials, because the nitrato groups are excellent chelating capping ligands in Ln(III) chemistry. A variety of Ln(NO_3_)_3_∙*n*H_2_O/LH (Ln = Gd-Yb) reactions systems, involving various solvent media, reagent ratios, absence/presence of external base and crystallization methods were systematically studied before establishing the optimized experiments described in Section 3. The only solvent that gave satisfactory crystallinity was methanol. In many instances, we isolated microcrystalline powders with acceptable analytical data, but we report here only the structurally characterized products which are Gd(III), Tb(III), Dy(III), and Er(III) complexes.

The 1:1 reactions between Ln(NO_3_)_3_∙*n*H_2_O (*n* = 5, 6) and HL in MeOH gave pale yellow solutions from which yellowish crystals of [Gd_2_(NO_3_)_4_(L)_2_(H_2_O)]∙2MeOH∙2H_2_O (**1**∙2MeOH∙2H_2_O), [Tb_2_(NO_3_)_4_(L)_2_(H_2_O)]∙2MeOH∙1.5H_2_O (**2**∙2MeOH∙1.5H_2_O), [Dy_2_(NO_3_)_4_(L)_2_(MeOH)_0.7_(H_2_O)_0.3_]∙2.5MeOH (**3**∙2.5MeOH) and [Er_2_(NO_3_)_4_(L)_2_(H_2_O)]∙3MeOH∙0.5H_2_O (**4**∙3MeOH∙0.5H_2_O) were subsequently isolated in moderate to good yields. Unit-cell determination of poor-quality crystals of the Eu(III) complex proved that this is most probably isomorphous to **1**∙2MeOH∙2H_2_O. Assuming that the complexes are the only products from the reaction system, their formation can be summarized by Equation (1); S is H_2_O for the Gd(III), Tb(III) and Er(III) complexes, and MeOH/H_2_O for the Dy(III) compound (vide infra). Use of bases, e.g., Et_3_N (Ln^III^:ligand:base = 1:1:1), gave the same complexes (by analytical data and IR evidences) in a powder form, Equation (2). The dried samples were used for the IR and UV/Vis spectra, whereas the as-isolated crystals were used for the magnetic measurements.
(1)$${2\;\mathrm{Ln}(\mathrm{NO}}_{3}{)}_{3}\cdot n{\;H}_{2}O+2\;\mathrm{HL}+S\;\overset{\mathrm{MeOH}}{\to }{\;[\mathrm{Ln}}_{2}{{(\mathrm{NO}}_{3})}_{4}{\left(L\right)}_{2}{(S)]+2\;\mathrm{HNO}}_{3}+2n{\;H}_{2}O$$
(2)$${2\;\mathrm{Ln}(\mathrm{NO}}_{3}{)}_{3}\cdot n{\;H}_{2}O+2\;\mathrm{HL}+2\;B+S\;\overset{\mathrm{MeOH}}{\to }{\;[\mathrm{Ln}}_{2}{{(\mathrm{NO}}_{3})}_{4}{\left(L\right)}_{2}{(S)]+2\;(\mathrm{BH})(\mathrm{NO}}_{3})+2n{\;H}_{2}O$$

When the reactions are performed as illustrated in Equation (1) with La(III), Ce(III), Pr(III) and Nd(III), single crystals of [Ln(NO_3_)_3_(HL)(MeOH)_2_] (Ln = La, Ce) [50], [Nd(NO_3_)_3_(HL)(H_2_O)]∙MeOH [50] and [Pr(NO_3_)_3_(HL)(H_2_O)]∙MeOH were obtained. The identity of the La(III), Ce(III) and Nd(III) complexes was confirmed by the unit cell comparisons with those of the reported materials [50], while the unit cell dimensions of the Pr(III) compound (which was not reported by Xu et al.) are very similar to those of its Nd(III) counterpart. We attribute the difference in the identity of the La(III)-Nd(III) and Eu(III)-Dy(III), Er(III) complexes to the smaller polarizing ability of the lighter and larger Ln^III^ ions, whose weaker coordination to the oxygen atom of the enol-imino form of LH in solution is not enough to remove the acidic hydrogen atom from the -OH group. In contrast to what was reported in the literature [50], when Et_3_N is used in the reaction mixtures involving La(III), Ce(III), Pr(III) and Nd(III) ions with HL in MeOH (i.e., reactions analogous to those described by Equation (2)), powders of uncertain nature are isolated. Their characterization could not be proceeded further; their IR spectra, however, resemble those of **1**–**4** suggesting deprotonation of the ligand.

In the IR spectra of the well-dried samples **1**–**4**, the broad band centered at ∼3420 cm^−1^ is due to the ν(OH) vibration of coordinated H_2_O (**1**–**4**) or MeOH (**3**) [48], while the δ(OH) vibrations appear at ∼1615 cm^−1^. The medium-intensity band at 3316 cm^−1^ and the very strong band at 1702 cm^−1^ in the spectrum of the free ligand HL are assigned to the ν(NH) and ν(C=O) vibrations, respectively [47,54]. These bands suggest that HL, whose crystal structure is not known, exists in its keto-amino form in the solid state, and not in the enol-imino form shown in Figure 1. There are no bands in the regions 3400–3100 and 1720–1625 cm^−1^ regions of the spectra for **1**–**4**, indicating that the carbon-oxygen bond of coordinated L¯ does not have an appreciable double bond character [67]. This conclusion is also supported by the single-crystal X-ray structures of the complexes (vide infra). The highest wavenumber bands in the 1600–400 cm^−1^ region are at 1594 (**1–3**) and 1596 cm^−1^ (**4**), assigned to a pyridyl stretching vibration [48]. The bands at ∼1470 and ∼1280 cm^−1^ are assigned [68,69] to the ν_1_(A_1_)[ν(N=O)] and ν_5_(Β_2_)[ν_as_(NO_2_) modes, respectively, of the coordinated nitrato group. The separation of these two, highest-frequency stretching bands is large (∼190 cm^−1^), indicating the bidentate character of the nitrato ligands [68]. The IR spectra of the crystalline samples (i.e., those containing lattice solvent molecules) are very similar with those of the fully dried samples in the 3500–2800 cm^−1^ region and almost identical in the 1700–400 cm^−1^ region.

The solid-state (diffuse reflectance) spectra of **2**–**4** (Figure 3) are dominated by broad intraligand bands in the 280–430 nm range. The f-f transitions could be seen, and these have been assigned on the basis of the well characterized energy-level diagrams [70,71,72] for the Ln^III^ ions concerned [70,71,72]. Specific assignments are as follows (the wavelengths are in parentheses): Complex **2**: ^7^*F*_6_→^7^*F*_3_ (1987 nm); ^7^*F*_6_→^7^*F*_2_ (1915 nm); ^7^*F*_6_→^7^*F*_1_ (1848 nm); ^7^*F*_6_→^7^*F*_0_ (1654 nm), ^7^*F*_6_→^5^*D*_4_ (490 nm, sh). Complex **3**: ^6^*H*_15/2_→^6^*H*_11/2_ (1702 nm); ^6^*H*_15/2_→^6^*H*_9/2_, ^6^*F*_11/2_ (1301 nm); ^6^*H*_15/2_→^6^*F*_9/2_, ^6^*H*_7/2_ (1100 nm); ^6^*H*_15/2_→^6^*H*_5/2_ (975 nm); ^6^*H*_15/2_→^6^*F*_7/2_ (910 nm); ^6^*H*_15/2_→^6^*F*_5/2_ (811 nm); ^6^*H*_15/2_→^6^*F*_3/2_ (757 nm); ^6^*H*_15/2_→^6^*F*_1/2_ (738 nm, sh); ^6^*H*_15/2_→^4^*F*_9/2_ (480 nm, sh). Complex **4**: ^4^*I*_11/2_→^4^*F*_9/2_ (1917 nm); ^4^*I*_15/2_→^4^*I*_13/2_ (1488 nm); ^4^*I*_15/2_→^4^*I*_11/2_ (960 nm); ^4^*I*_15/2_→^4^*I*_9/2_ (802 nm); ^4^*I*_15/2_→^4^*F*_9/2_ (648 nm); ^4^*I*_15/2_→^4^*S*_3/2_ (538 nm); ^4^*I*_15/2_→^2^*H*_11/2_ (518 nm); ^4^*I*_15/2_→^4^*F*_7/2_ (483 nm). The unusually high intensity of the band assigned to the ^4^*I*_15/2_→^2^*H*_11/2_ transition in the spectrum of **4** is due to the fact that this transition is “hypersensitive” [72,73]; the absence of splitting of this band might indicate that the coordination number of Er(III) is higher than 8 in this compound [65,72,73]. The diffuse reflectance spectra of the crystalline samples (i.e., those containing lattice solvent molecules) are identical with those of the fully dried samples.

### 2.2. Description of Structures

The structures of **1**∙2MeOH∙2H_2_O, **2**∙2MeOH∙1.5H_2_O, **3**∙2.5MeOH, and **4**∙3MeOH∙0.5H_2_O were determined by single-crystal X-ray crystallography. Crystallographic data are listed in Table 1. Various structural plots are shown in Figure 4, Figure 5 and Figure 6 and Appendix A. Selected interatomic distances and bond angles are given in Table 2.

As shown by Figure 4 and Figure 5 and Appendix A, the four complexes have similar molecular structures. The main difference is that the Gd(III), Tb(III), and Er(III) complexes contain one coordinated aqua ligand that is replaced by a mixture of coordinated H_2_O and MeOH molecules in the Dy(III) complex, with a ca. 30%: 70% ratio (±15%). This ratio has been observed on two different crystals. Despite very close metrics for the unit cells (Table 1), the four complexes differ slightly by their lattice solvent contents. A general description of the molecular structures is given below.

In the dinuclear molecules, the two Ln^III^ ions are doubly bridged by the deprotonated oxygen atoms (O1, O2) of the two parallel (or “head-to-head”) 2.21011 ligands (Figure 2). The Ln^III^∙∙∙Ln^III^ distances fall in the 4.000(1)–3.933(1) Å range, while the Ln-O-Ln bond angles are in the narrow 113.7(2)–115.4(1)° range. The central {Ln_2_(μ-OR)_2_}^4+^ core appears to be nearly rhombic. Two pyridyl nitrogen (N1, N5) and two “imino” nitrogen atoms (N2, N6), as well as two slightly anisobidentate chelating nitrato groups (O3/O4, O6/O8) complete 10-coordination at Ln1 creating a {Ln1O_6_N_4_} coordination environment. The {Ln2O_7_N_2_} 9-coordinate environment at Ln2 is completed by two pyridyl nitrogen atoms (N4, N8), two almost symmetrical bidentate chelating nitrato groups (O9/O11, O12/O14) and one terminal aquo (for **1**∙2MeOH∙2H_2_O, **2**∙2MeOH∙1.5H_2_O and **4**∙3MeOH∙0.5H_2_O) or methanol (**3**∙2.5MeOH) ligands (the donor atom is O15 or O1M). For a given bond, the Ln-O/N bond lengths generally follow the Gd > Tb > Dy > Er order due to lanthanide contraction; this trend is also observed for the Ln∙∙∙Ln distances. Establishing the site of deprotonation is often difficult for L¯ and related polydentate ligands [48], but the C8-O1 (average 1.305 Å), N3-C8 (average 1.294 Å), N2-N3 (average 1.408 Å), C6-N2 (average 1.296 Å) and C21-O2 (average 1.309 Å), N7-C21 (average 1.304 Å), N6-N7 (average 1.396 Å), C19-N6 (average 1.297 Å) distances indicate a charge delocalization within the OCNNC backbone of the two deprotonated ligands. If the deprotonation must be localized on a specific atom, the relatively long C8-O1 (average 1.305 Å) and C21-O2 (average 1.309 Å) distances, which are not typical of a coordinated carbonyl group, let us believe that the O atoms are the principal sites of deprotonation.

To evaluate in more detail the coordination polyhedra defined by the donor atoms around Ln1 and Ln2, the experimental structural data were compared to the theoretical values for the most common polyhedra with ten and nine vertices using SHAPE [74]. The so-named continuous shape measures (CShM) approach allows researchers to numerically estimate how far an experimental coordination sphere of a metal ion deviates from an ideal polyhedron. Of the accessible 10-coordinate polyhedra for metal centers, the sphenocorona (tetradecahedron) is the most appropriate for the description of the 10 donor atoms around Ln1 in the four compounds. The best fit for the Ln2 ions was obtained for the spherical capped square antiprism, with the nitrato oxygen atom O11 being the capping atom. Since the nitrato groups impose a small bite angle, the polyhedra of the Ln^III^ centers in the four compounds are distorted [69]. The polyhedra of Dy1 and Dy2 in **3**∙2.5MeOH are shown in Figure 6, while numerical data are listed in Appendix A.

In the crystal structures, the lattice and coordinated H_2_O and MeOH molecules are *H*-bonded connecting the complexes into 3D
*H*-bonded networks for **1**∙2MeOH∙2H_2_O and **2**∙2MeOH∙1.5H_2_O, and an 1D
*H*-bonded network for **4**∙3MeOH∙0.5H_2_O. Because the solvents are highly disordered with partial occupancies for **3**∙2.5MeOH (vide infra), its exact dimensionality remains undetermined (1D or higher).

Complexes **1**–**4** join the family of crystallographically characterized metal complexes based on HL or L¯ as ligands. The members of this family are listed in Table 3 along with information about the coordination mode of the ligands, the nuclearity and dimensionality of the compounds, and the coordination polyhedra of the metal centers involved. The 2.21011 ligation mode observed for the present dinuclear Ln(III) complexes has been identified in Mn(II), Ni(II) and Cu(II) (2 × 2) grids [47,48,54,55] and in the pentanuclear Mn(II) complex [Mn_5_(L)_6_](ClO_4_)_4_ [54], in which the metal ions define a trigonal bipyramid. Totally, six different coordination modes of the anionic L¯ ligand have been confirmed (Figure 2, Table 3), suggesting that this polydentate ligand is quite flexible; these results imply that, in addition to the criteria discussed in Section 1, serendipity (i.e., the “head-to-head” disposition of the two L¯ ligands) played also a role in the formation of asymmetric Ln^III^_2_ species.

### 2.3. Dc Magnetic Susceptibility Studies

Direct-current (dc) magnetic susceptibility (*χ*) data on polycrystalline samples of **1**∙2MeOH∙2H_2_O, **2**∙2MeOH∙1.5H_2_O, **3**∙2.5MeOH and **4**∙3MeOH∙0.5H_2_O were collected in the 1.8–300 K range using an applied field of 0.1 T. Magnetization (*M*) vs. magnetic field (*H*) at various low temperatures and vs. *HT*^−1^ (*T* is the absolute temperature) measurements were also performed. These magnetic data are shown in Figure 7 and Appendix A. For all complexes, the observed paramagnetism arises solely from the 4f Ln^III^ ions. At room temperature, the *χΤ* values of **1**∙2MeOH∙2H_2_O, **2**∙2MeOH∙1.5H_2_O, **3**∙MeOH∙2H_2_O and **4**∙3MeOH∙0.5H_2_O are 15.9, 23.0, 27.7 and 22.5 cm^3^·K·mol^−1^, respectively. These values are in very good agreement with the expected theoretical values (15.75, 23.63, 28.34, 22.96 cm^3^·K·mol^−1^, respectively) for two noninteracting Ln^III^ centers: Gd^III^ (^8^*S*_7/2_, *S* = 7/2, *L* = 0, *g* = 2), Tb^III^ (^7^*F*_6_, *S* = 3, *L* = 3, *g_J_* = 3/2), Dy^III^ (^6^*H*_15/2_, *S* = 5/2, *L* = 5, *g_J_* = 4/3) and Er^III^ (^4^*I*_15/2_, *S* = 3/2, *L* = 6, *g* = 6/5).

The presence/absence of magnetic exchange interaction between the two Ln^III^ and its strength in the dinuclear complexes can be revealed looking at the magnetic properties of the Gd(III) complex **1**, as the Gd^III^ centers present no spin-orbit coupling at the first order. The decrease of the *χT* product below ∼30 K (the value of the product is 14.0 cm^3^·K·mol^−1^ at 1.85 K) reveals directly the presence of antiferromagnetic Gd^III^∙∙∙Gd^III^ exchange interactions. The experimental data were fitted to the van Vleck analytical expression of the susceptibility derived from the following isotropic Heisenberg spin Hamiltonian: *H* = −2*J*(S_Gd1_∙S_Gd2_). The best-fit parameters obtained are *J*/*k*_B_ = −0.020(6) K and *g* = 2.023 (1). The observed antiferromagnetic exchange interaction is very weak as a consequence of the shielded 4f orbitals that have little overlap with the orbitals of the bridging oxygen atoms. Such very small *J* values have been observed in systems containing symmetrically-bridged {Gd^III^_2_(μ_2_-OR)_2_} cores [57,59,62,75,76]. However, the low symmetry of the Gd^III^ ion in **1** might give rise to small zero-field splitting and this, in turn, can also lead to the decrease of the *χT* product at low temperatures that would be phenomenologically included the *J* estimation [75].

Magnetization measurements at 1.85 K (Appendix A) reveal a saturation of 14.1 μ_Β_ at 7 T, in very good agreement with the expected theoretical value of 14.0 μ_Β_ for two *S* = 7/2 spins (*g* = 2.00). Indeed, as the interdimer antiferromagnetic interaction is very weak, the magnetization data below 8 K are perfectly fitted to the sum of two *S* = 7/2 Brillouin functions with *g* = 2.03(3) (Appendix A).

For the Tb(III), Dy(III) and Er(III) analogues, the *χT* product decreases upon cooling of the sample (Figure 7). The values of the *χT* product at 1.85 K and 0.1 T are 16.4, 20.1 and 11.2 cm^3^∙K∙mol^−1^ for **2**∙2MeOH∙1.5H_2_O, **3**∙2.5MeOH and **4**∙3MeOH∙0.5H_2_O, respectively. For these Ln^III^ ions with an unquenched orbital moment associated with a ligand field, there are three possible contributions to the observed behavior [75]: (a) the thermal depopulations of the Stark sublevels; (b) antiferromagnetic exchange interactions between the magnetic Ln^III^ centers; and (c) the presence of significant magnetic anisotropy. The magnetization values at 1.85 K and 7 T are 10.1, 10.6 and 9.8 μ_B_ for **2**∙2MeOH∙1.5H_2_O, **3**∙2.5MeOH and **4**∙3MeOH∙0.5H_2_O, respectively (Appendix A). The non-saturation of the magnetization indicates the presence of significant magnetic anisotropy and/or low-lying excited states in the three complexes [43,44]. This can be further confirmed by the lack of superposition and high-field variation on a single master curve of each *M* vs. *H*/*T* plot (Appendix A).

### 2.4. Ac Magnetic Susceptibilty Studies

In order to investigate the presence/absence of slow paramagnetic relaxation, in other words the SMM properties of these complexes, the dynamics of **2**∙2MeOH∙1.5H_2_O, **3**∙2.5MeOH and **4**∙3MeOH∙0.5H_2_O was probed by ac susceptometry (Figure 8, Figure 9 and Figure 10, Appendix A). In the zero dc field, an out-of-phase component (*χ*″) of the ac susceptibility was not detected above 1.8 K for ac frequencies up to 10 KHz. This is also true in presence of external dc fields up to 3 T for **2**∙2MeOH∙1.5H_2_O. However, in an applied dc field, clear frequency-dependent in-phase (*χ*′) and out-of-phase signals were detected for **3**∙2.5MeOH and **4**∙3MeOH∙0.5H_2_O (Appendix A). The appearance and the slowing down of the relaxation when applying a small dc field is expected when the magnetization relaxation is dominated by quantum tunneling (quantum tunneling of magnetization, QTM) [3,77] and/or Raman processes [78]. In the QTM case, the application of a dc field minimizes the probability of the magnetization to tunnel, as already observed in many Ln(III) SMMs [18,19,23,34,57,69,79]. The temperature and frequency dependencies of the ac susceptibility were thus studied under an optimum dc field (i.e., a dc field for which the relaxation process of the magnetization exhibits a good compromise between its characteristic time and the intensity of the relaxation mode) of 600 Oe for **3**∙2.5MeOH and 1000 Oe for **4**∙3MeOH∙0.5H_2_O (Figure 8 and Figure 9).

The combination of all these frequency dependent ac data (Figure 8 and Figure 9, Appendix A) and their fit to the generalized Debye model [80] allows the determination of the field and temperature dependence of the relaxation time, τ, and its estimated standard deviation (ESD) [81], which quantifies the broadness of the relaxation time distribution. As shown in Figure 10, the obtained ESDs are significantly large and thus the following discussion of the relaxation time must be taken with a certain caution.

Paramagnetic relaxation [82] usually has four principal origins, which include Raman [78], direct [82], thermally activated (Arrhenius) [82], and QTM [3,77,83,84] relaxation processes, as summarized in the following Equations (3) and (4) [83]:(3)$$\tau^{-1}=\tau_{Raman}^{-1}+\tau_{Direct}^{-1}+\tau_{Arrhenius}^{-1}+\tau_{QTM}^{-1}$$
(4)$$\tau^{-1}=C\frac{1+C_{1}H^{2}}{1+C_{2}H^{2}}T^{n}+ATH^{4}+\tau_{0}^{-1}\exp\left(-\frac{\Delta }{k_{B}T}\right)+\frac{B_{1}}{1+B_{2}H^{2}}$$

As illustrated by these equations, each relaxation mechanism has its own characteristic temperature (*T*) and field (*H*) dependence. The field dependence of the relaxation time, shown on the left part of Figure 10, possesses a similar variation for **3**∙2.5MeOH and **4**∙3MeOH∙0.5H_2_O. At low fields, an increase of the relaxation time is first observed, in agreement with only QTM or Raman (with *C*_2_ > *C*_1_) pathways, while the *H*^4^ variation at higher fields strongly suggests a Direct relaxation mechanism. Therefore, to model the experimental field dependence of the relaxation time, two simple approaches have been used considering (i) Raman and direct, and (ii) QTM and direct processes.

Both models are able to reproduce the experimental τ vs. *H* data, but only the first one, when an Orbach relaxation is introduced, is able to fit both τ vs. *H* and τ vs. *T* variations as shown in Figure 10. The best parameters of the three first terms of Equation (4) are for **3**∙2.5MeOH: *C*_1_ = 25(5) T^−2^, *C*_2_ = 396(50) T^−2^, *C* = 88(5) s^−1^·K^−3.4^, *n* = 3.4(1), *A* = 2.9(4) × 10^5^ K^−1^·T^−4^·s^−1^, *τ*_0_ = 6.3(5) × 10^−8^ s and ∆_eff_/*k*_B_ =34(2) K; and for **4**∙3MeOH∙0.5H_2_O: *C*_1_ = 222(20) T^−2^, *C*_2_ = 461(50) T^−2^, *C* = 652(50) s^−1^·K^−4.8^, *n* = 4.8(5), *A* = 1.2(7) × 10^6^ K^−1^·T^−4^·s^−1^, *τ*_0_ = 5.6(5) × 10^−11^ s and ∆_eff_/*k*_B_ = 39(2) K. Considering the necessity of three relaxation mechanisms with seven adjustable parameters in the above model, alternative descriptions are likely possible, even if we were not able to find another model involving only three relaxation processes. In particular, the origin of an energy gap of 34 K (for **3**∙2.5MeOH) and 39 K (for **4**∙3MeOH∙0.5H_2_O) for these Dy(III) and Er(III) complexes is not obvious and should certainly be questioned in the present model. To conclude this analysis, keeping in mind that it should be taken with great caution as indicated above, these results suggests that Raman, direct and an Orbach-like processes are at the origin of the observed SMM properties in **3**∙2.5MeOH and **4**∙3MeOH∙0.5H_2_O.

At this stage, we are doing a brief parenthesis to mention (for the non-familiar readers) that in a Direct relaxation process [82], a transition between one *m_j_* state to a different one is by emission/absorption of a phonon to/from the surrounding bath, with the same quanta of energy of the transition. This mechanism is efficient at very low temperatures only under an applied magnetic field and it is strongly sample- and Ln^III^- dependent. The Raman relaxation pathway [78] involves the inelastic scattering of phonons, where the energies of the involved phonons decrease or increase, with the difference in energy (*h**ν*_i+1_ − *h**ν*_i_) being absorbed or released by the system. When the scattered energy causes a direct transition, the respective relaxation process is named first-order Raman, whereas spin relaxation occurring via virtual states is termed second-order Raman mechanism. The Raman pathways require the presence of phonons and become important at finite temperatures. The Raman relaxation mechanisms depend also strongly on the sample and its composition. In the Orbach process [82], the absorption of a phonon by the spin system causes an excitation to a low-lying excited state. This is accompanied by the emission of a phonon with an energy corresponding to the difference between the ground and the low-lying excited state; this provides a pathway for the relaxation between the low-lying states. From the just mentioned processes, only the Orbach one leads to a thermally activated dependence. It is evident that all three magnetization relaxation mechanisms are spin-lattice relaxation processes [82,85].

In an effort to understand the absence of SMM properties at zero field for **3**∙2.5MeOH, we have determined the orientation of the ground-state magnetic anisotropy axes for the Dy^III^ centers in the complex (Appendix A), using a method [25] resulting from an electrostatic model that requires only the knowledge of the single-crystal structure of the complex. Definite conclusions based on the orientations of the anisotropy axes would be risky. Nevertheless for Dy2, the easy axis does not point towards the “axial” O11 atom, which is the capping atom of the capped square antiprismatic polyhedron of this metal ion. This orientation forces the oblate f-electron density [24] of Dy2 to be roughly parallel to the easy axis, which is a non-favorable configuration to achieve significant slow magnetization relaxation [25,69]. However, the distribution of the charged oxygen atoms (and the derived field) is rather spherical for both metal ions and any oblate-prolate [24] discussions are perhaps meaningless in the present case.

## 3. Experimental Section

### 3.1. Materials, Physical and Spectroscopic Measurements

All manipulations were performed in the normal laboratory atmosphere using reagents and solvents (Alfa Aesar, Aldrich; Karlsruhe, Germany and Tanfrichen, Germany, respectively) as received. The organic ligand LH was synthesized in typical yields of >85% as described in the literature [47,49,51,54], i.e., by the 1:1 reaction between picolinic acid hydrazide and 2-acetylpyridine in refluxing EtOH for 3 h. Its purity was checked by microanalyses (C, H, N), determination of the melting point (found, 192–193 °C; reported 195–197 °C), and IR and ^1^H NMR spectra. Elemental analyses (C, H, N) were performed by the University of Patras (Patras, Greece) Center of Instrumental Analysis. FT-IR spectra (4000–400 cm^−1^) were recorded using a Perkin-Elmer (Perkin-Elmer, Watham, MA, USA) 16PC spectrometer with samples prepared as KBr pellets. Solid-state electronic (diffuse reflectance) spectra were obtained with an Agilent Cary 5000 instrument (Agilent Technologies, Santa Clara, CA, USA) in the 250–2000 nm range by using an integrating sphere. A KBr reference pellet was prepared by grinding pure KBr (CAS number 7758-02-3, Sigma Aldrich, Athens, Greece, purity >99%) in a pestle and mortar until a homogenous fine powder was obtained. This was then placed in a pellet-forming die under a ∼8 tons pressure for about 5 min until a homogenous disc was formed. Discs of the samples of the compounds, mixed and pressed with KBr (roughly 2% in KBr), were prepared in a similar manner. Magnetic measurements were performed on a Quantum Design SQUID magnetometer MPMS-XL (Quantum Design, San Diego, CA, USA) and a PPMS-II susceptometer, housed at the Centre de Recherche Paul Pascal at temperatures between 1.8 and 300 K, and dc magnetic fields ranging from –7 to +7 T. ac magnetic susceptibility measurements were performed in an oscillating ac field of 1 to 6 Oe with frequencies between 10 and 10,000 Hz. The measurements were carried out on polycrystalline samples received directly from the mother liquors (16.70 mg for **1**∙2MeOH∙2H_2_O, 13.70 mg for **2**∙2MeOH∙1.5H_2_O, 11.76 mg for **3**∙2.5MeOH and 7.51 mg for **4**∙3MeOH∙0.5H_2_O), suspended in mineral oil (typically 10–30 mg) and introduced in a sealed polypropylene bag (3 × 0.5 × 0.02 cm; typically 7–21 mg). Prior to the main experiments, the field-dependent magnetization was measured at 100 K on each sample in order to detect the possible presence of any bulk ferromagnetic impurities. In fact, paramagnetic materials should exhibit a perfect linear dependence of magnetization that extrapolates to zero at zero dc field; all samples appeared to be free of any bulk ferromagnetic impurities. The magnetic data were corrected for the sample holder, the oil and the intrinsic diamagnetic contributions.

### 3.2. Synthesis of the Representative Complex [Gd_2_(NO_3_)_4_(L)_2_(H_2_O)]∙2MeOH∙2H_2_O (**1**∙2MeOH∙2H_2_O)

To a stirred colorless solution of HL (0.024 g, 1.00 mmol) in MeOH (12 mL), solid Gd(NO_3_)_3_∙6H_2_O (0.045 g, 1.00 mmol) was added. The solid soon dissolved and the resulting pale yellow solution was stirred for a further 5 min, filtered and stored in a closed flask at room temperature. X-ray quality, yellow crystals of the product were obtained within a period of one week. The crystals were collected by filtration, washed with cold MeOH (2 × 0.5 mL) and Et_2_O (4 × 2 Ml,) and dried in a vacuum desiccator over P_4_O_10_ for 24 h. Yield: 42%. The complex was satisfactorily analyzed as lattice solvent-free, i.e., as **1**. Anal. Calcd. (%) for C_26_H_24_N_12_Gd_2_O_15_: C, 29.48; H, 2.29; N, 15.87. Found (%): C, 29.71; H, 2.24; N, 15.63. IR (KBr, cm^−1^): 3420mb, 3040w, 3005w, 2920w, 1616m, 1594m, 1576m, 1555sh, 1544m, 1506sh, 1472s, 1445sh, 1358s, 1280sb, 1252sh, 1164m, 1136w, 1100w, 1026m, 1012m, 998w, 920m, 816m, 808sh, 788m, 742m, 708m, 698m, 676m, 636m, 576w, 558w, 518w, 486w, 468m, 418w, 402w.

### 3.3. Syntheses of the Complexes [Τb_2_(ΝO_3_)_4_(L)_2_(H_2_O)]∙2ΜeOH∙1.5H_2_O (**2**∙2ΜeOH∙1.5H_2_O), [Dy_2_(ΝO_3_)_4_(L)_2_(MeOH)]∙ΜeOH∙2H_2_O (**3**∙2.5ΜeOH) and [Er_2_(ΝO_3_)_4_(L)_2_(H_2_O)]∙3ΜeOH∙0.5H_2_O (**4**∙3MeOH∙0.5H_2_O)

These complexes were prepared in an identical manner with **1**∙2MeOH∙2H_2_O by simply replacing Gd(NO_3_)_3_∙6H_2_O with the equivalent amount of the appropriate Ln(NO_3_)_3_∙*n*H_2_O (*n* = 5 or 6). The yields were 57, 61 and 48% for the Tb(III), Dy(III) and Er(III) complexes, respectively. The complexes were satisfactorily analyzed as lattice solvent-free, i.e., as **2**, **3**, and **4**. In some samples of the Tb(III) and Dy(III) complexes, a small percentage of lattice H_2_O (typically 0.2–0.4 moles per mole of the complex) could also fit the experimental analytical data. The cited experimental values here are for samples found dried from a TGA experiment. Anal. Calcd. (%) for C_26_H_24_N_12_Tb_2_O_15_ (**2**): C, 29.39; H, 2.28; N, 15.82. Found (%): C, 29.21; H, 2.31; N, 15.63. Anal. Calcd. (%) for C_27_H_26_Ν_12_Dy_2_O_15_ (**3**): C, 29.92; H, 2.42; Ν, 15.51. Found (%): C, 30.06; H, 2.38; Ν, 15.40. Anal. Calcd. For C_26_H_24_Ν_12_Εr_2_O_15_ (**4**): C, 28.94; H, 2.25; Ν, 15.58. Found (%): C, 28.83; H, 2.30; N, 15.32. The IR spectra of these complexes are almost superimposable with the spectrum of **1** with a maximum wavenumber difference of ±4 cm^−1^.

### 3.4. Single-Crystal X-ray Crystallography

Crystallographic data (Table 1) were collected with a Bruker APEX II Quasar diffractometer (Bruker Scientific LLC, Billerica, MA, USA), equipped with a graphite monochromator centered on the path of Mo Kα radiation. Single crystals of the complexes, obtained as described in Paragraphs 3.2 and 3.3, were coated with Cargille^TM^ NHV immersion oil and mounted on a fiber loop, followed by data collection at 120 or 230 K. The program SAINT was used to integrate the data, which was thereafter corrected using SADABS [86]. The structures were solved using direct methods and refined by a fully-matrix least-squares method on *F*^2^ using SHELXL-2014 [87]. All non-H atoms were refined with anisotropic displacement parameters. Some details are listed below.

For **1**∙2MeOH∙2H_2_O: H atoms were assigned to ideal positions and refined isotropically using a riding model, except those of the H_2_O and MeOH molecules for which the positions of H atoms were found on the difference Fourier map and refined using positional constraints.

For **2**∙2ΜeOH∙1.5H_2_O: H atoms were assigned to ideal positions and refined isotropically using a riding model, except those of the coordinated H_2_O molecule for which the positions of H atoms were introduced based on the different Fourier map. H atoms on some lattice MeOH and H_2_O molecules were not introduced but were taken into account in the formula of the complex. Two crystallographically different MeOH molecules were refined with 0.5 occupancies, one of them being very close to a lattice H_2_O molecule with a 0.5 occupancy (O19W).

For **4**∙3MeOH∙0.5H_2_O: H atoms were assigned to ideal positions and refined isotropically using a riding model, except those of the coordinated H_2_O molecule for which the positions of H atoms were introduced based on the difference Fourier map. H atoms on some lattice MeOH and H_2_O molecules were not introduced but were taken into account in the formula of the complex. Three crystallographically different MeOH molecules were refined with 0.5 occupancies, one of them being very close to a H_2_O molecule with a 0.5 occupancy (O19W).

For **3**∙2.5ΜeOH: H atoms were assigned to ideal positions and refined isotropically, except those of the coordinated MeOH molecule for which the positions of H atoms were located on the difference Fourier map and refined using DFIX constraint. The relative electronic densities found on the C and O sites of this coordinated MeOH molecule suggest a cocrystallization of complexes with coordinated methanol and water. The ratio between coordinated MeOH and H_2_O was found to be ca. 70%: 30% (±15 %) and refined as this. The given formula is taking into account only the major component, i.e., with MeOH only. The crystal structure contains also a large void space that is filled by disordered lattice solvent molecules. As suggested by the interatomic distances, those were refined as lattice MeOH molecules using DFIX, EADP and SIMU constraints/restraints and with partial occupancies of some of them (1 for C2M/O2M, 0.6 for C3M/O3M disordered on two positions with equal occupancies, 0.5 for C4M/O4M disordered on 2 positions with equal occupancies, and 0.5 for C5M/O5M). H atoms were not introduced when involving disordered solvent molecules, but are taken into account in the compound formula. Due to the use of constraints and partial occupancies, the exact number and nature of those lattice solvent molecules should be taken with care.

Crystallographic data have been deposited with the Cambridge Crystallographic Data Center, No 2002796 (**1**∙2MeOH∙2H_2_O), 2002794 (**2**∙2ΜeOH∙1.5H_2_O), 2002797 (**3**∙2.5ΜeOH) and 2002795 (**4**∙3MeOH∙0.5H_2_O). Copies of the data can be obtained free of charge upon application to CCDC, 12 Union Road, Cambridge, CB2 1EZ, UK: Tel.: +(44)-1223-762910; Fax: +(44)-1223-336033; E-mail: deposit@ccdc.cam.ac.uk, or via http://www.ccdc.cam.ac.uk/conts/retrieving.html.

## 4. Concluding Comments and Perspectives

In this work, we have shown that, in addition to the mononuclear complexes [Ln(NO_3_)_3_(HL)(MeOH)] (Ln = La, Ce) and [Nd(NO_3_)_3_(HL)(H_2_O)] reported earlier [50] where the neutral keto-amino ligand LH behaves as a tridentate chelating, the monoanion L¯ can act as a bridging ligand forming asymmetric dinuclear complexes of the general formula [Ln_2_(NO_3_)_4_(L)_2_(S)] (S = H_2_O, MeOH). Four analogous complexes (Ln = Gd, Tb, Dy, Er) have been structurally characterized. Complexes **1**–**4** are new, welcome members of the growing group of complexes containing HL and L¯ (Table 3). This is the main chemical message from our efforts.

The important point of this work from a magnetic point of view is the absence of SMM behavior for the dinuclear Tb(III) complex, while the SMM properties of the Dy(III) and Er(III) analogues can be studied only when a magnetic field is applied. The rather spherical distribution of the charged oxygen atoms (which implies weak magnetic axiality around the oblate Tb^III^ and Dy^III^ centers and weak plane magnetic anisotropy around the prolate Er^III^ ions) is probably responsible for this. For the Dy(III) and Er(III) complexes, the detailed analysis of the relaxation time studying its thermal and field variations reveals the presence of multiple relaxation processes involving a direct relaxation and likely Raman and Orbach-like mechanisms.

We are currently investigating the possibility to prepare polynuclear Ln(III) complexes based on L¯, either “forcing” this ligand to act as triply bridging (μ_3_) or/and by using other potentially bridging ancillary ligands. Also, we have been working with bulkier analogues of HL, i.e., by replacing one or both 2-pyridyl groups by 2-quinolyl groups and by using a phenyl group instead of a methyl group, in an effort to reduce the coordination numbers of the Ln^III^ centers in asymmetric dinuclear complexes and achieve slow magnetic relaxation at zero field.

## Figures and Tables

**Figure 1 molecules-25-03153-f001:**
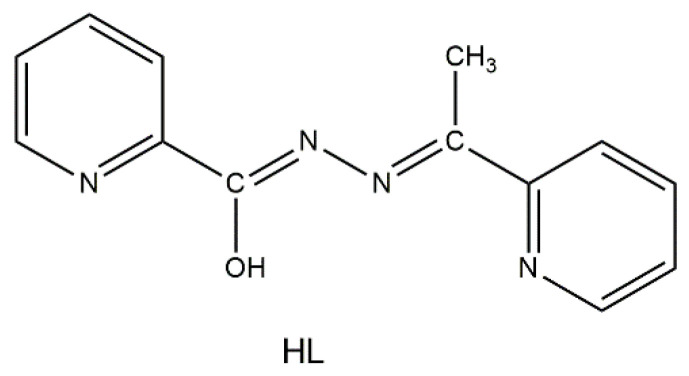
Structural formula of the free ligand *N*’-(1-(pyridin-2-yl)ethylidene)pyridine-2-carbohydrazide, drawn in its enol-imino tautomer, and its abbreviation.

**Figure 2 molecules-25-03153-f002:**
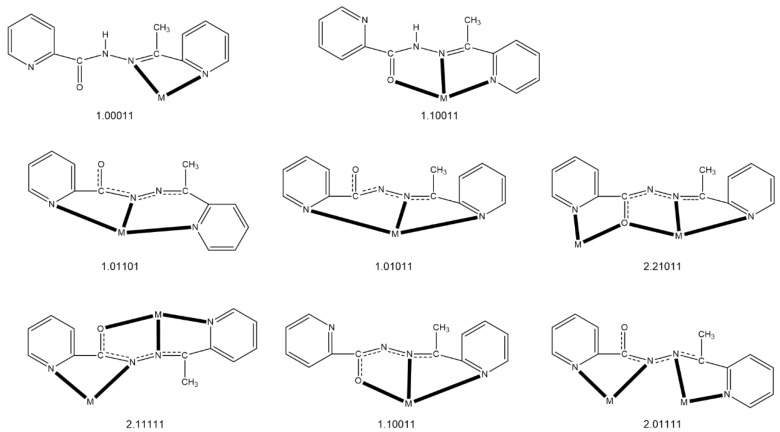
To date the crystallographically confirmed ligation modes of HL or L¯, and the Harris notation that describes these modes. The neutral ligand exists in the keto-amino form in the complexes. In the anionic ligand, the central OCNNC backbone has been drawn in a manner that emphasizes its delocalized nature which appears in most complexes. The coordination bonds are drawn with bold lines. M = metal ion. The structurally characterized complexes are listed in Table 3.

**Figure 3 molecules-25-03153-f003:**
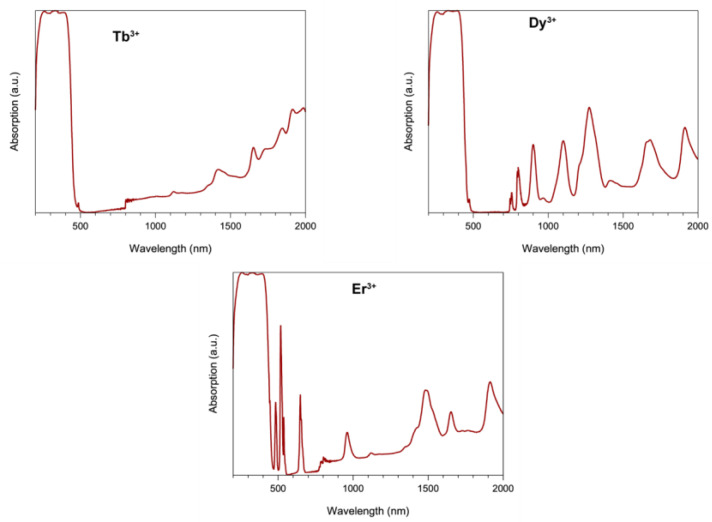
Solid-state (diffuse reflectance) electronic spectra of complexes **2** (**top left**), **3** (**top right**) and **4** (**bottom**) in the 250–2000 nm range.

**Figure 4 molecules-25-03153-f004:**
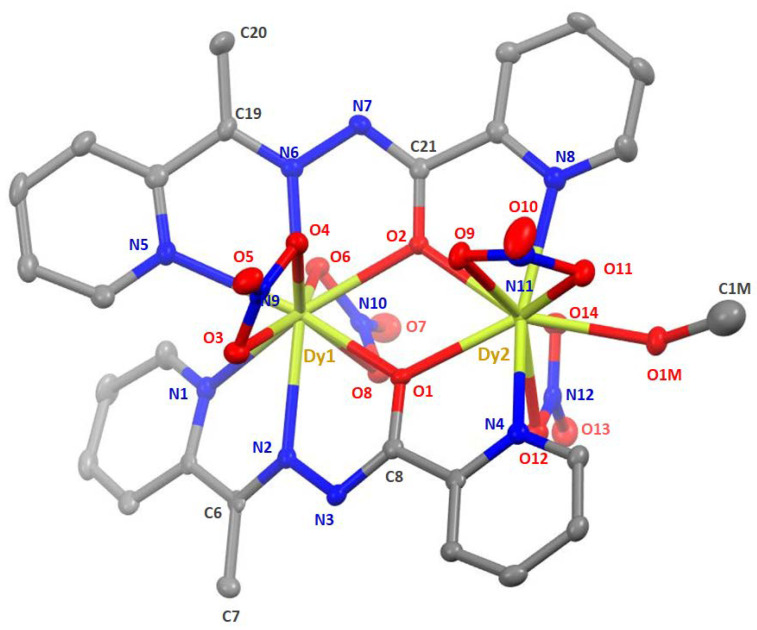
Molecular structure of [Dy_2_(NO_3_)_4_(L)_2_(MeOH)] as found in **3**∙2.5MeOH at 120 K. Thermal ellipsoids are depicted at 50% probability level. Hydrogen atoms are omitted for clarity. Note that only the major complex is depicted here, i.e., with coordinated MeOH instead of H_2_O, C1M having an occupancy of ca. 0.7.

**Figure 5 molecules-25-03153-f005:**
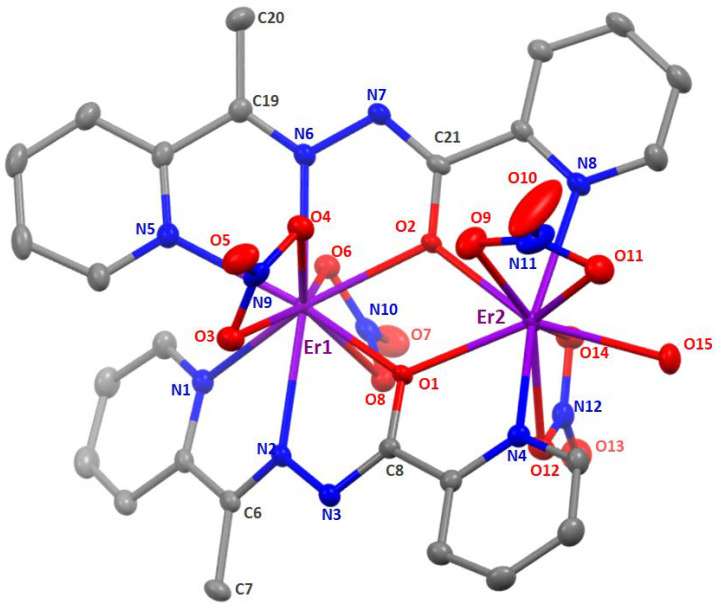
Molecular structure of [Er_2_(NO_3_)_4_(L)_2_(H_2_O)] as found in **4**∙3MeOH∙0.5H_2_O. Thermal ellipsoids are depicted at 50% probability level. Hydrogen atoms are omitted for clarity.

**Figure 6 molecules-25-03153-f006:**
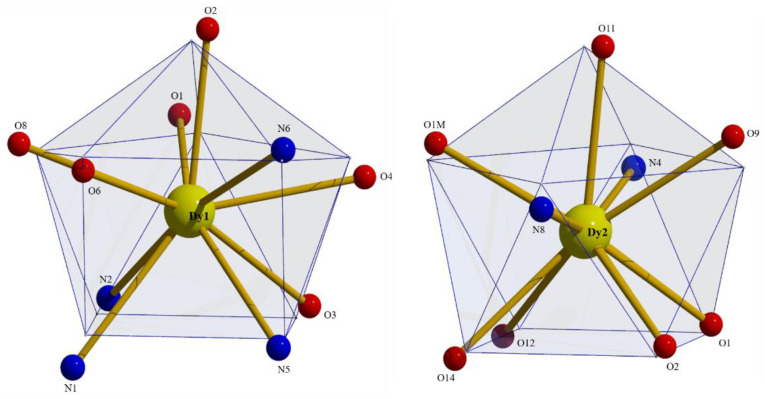
Sphenocoronal and spherical capped square antiprismatic coordination geometries of Dy1 and Dy2, respectively, in the structure of **3**∙2.5MeOH. The plotted polyhedra are the ideal, best-fit polyhedra using the program SHAPE [74].

**Figure 7 molecules-25-03153-f007:**
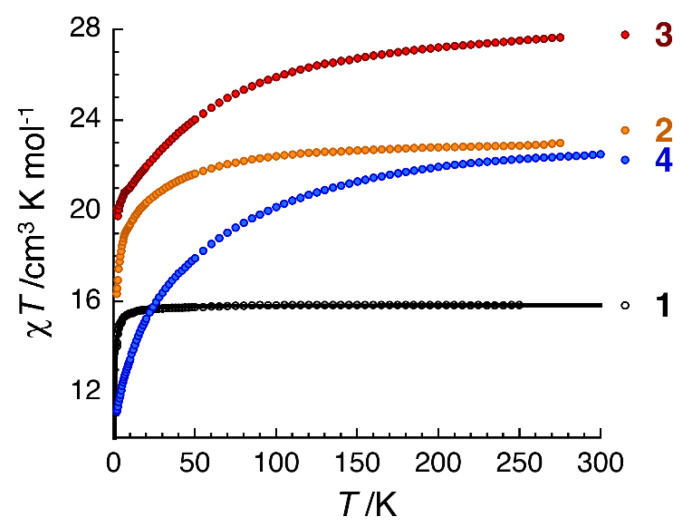
Temperature dependence of the *χT* product for the four complexes (**1**∙2MeOH∙2H_2_O, in black; **2**∙2MeOH∙1.5H_2_O in orange; **3**∙2.5MeOH in red and **4**∙3MeOH∙0.5H_2_O in blue) discussed in this paper at 0.1 T (*χ* is defined as *M*/*H* per mole of the respective complex). The solid black line is the fit of the data to the theoretical Heisenberg model for a dinuclear Gd^III^_2_ complex; see the text for details.

**Figure 8 molecules-25-03153-f008:**
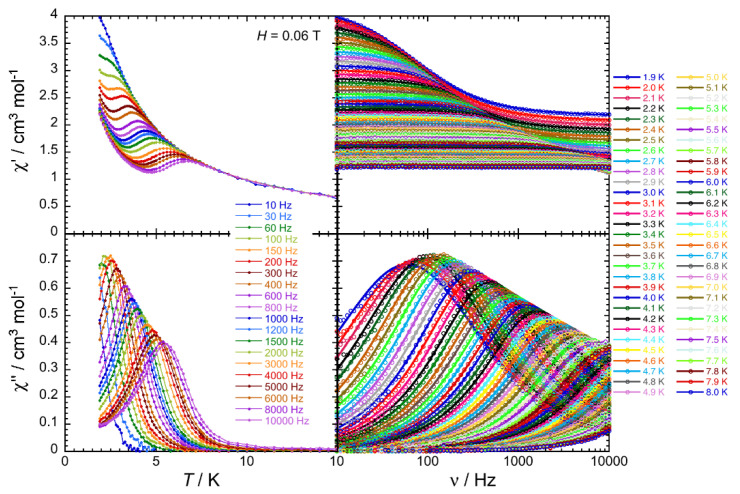
Frequency dependence of the real, in-phase (*χ*′, **top**) and imaginary, out-of-phase (*χ*″, **bottom**) components of the ac susceptibility under an external dc field of 600 Oe (0.06 T) at the indicated temperatures for complex **3**∙2.5MeOH. Solid lines are visual guides on the left plots, while they show the generalized Debye fit of the ac data on the right.

**Figure 9 molecules-25-03153-f009:**
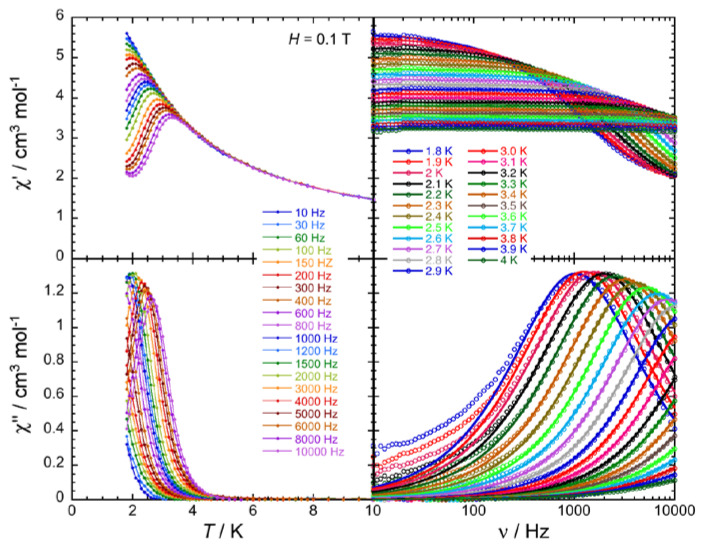
Frequency dependence of the real, in-phase (*χ*′, **top**) and imaginary, out-of-phase (*χ*″, **bottom**) components of the ac susceptibility under an external dc field of 1000 Oe (0.1 T) at the indicated temperatures for complex **4**∙3MeOH∙0.5H_2_O. Solid lines are visual guides on the left plots, while they show the generalized Debye fit of the ac data on the right.

**Figure 10 molecules-25-03153-f010:**
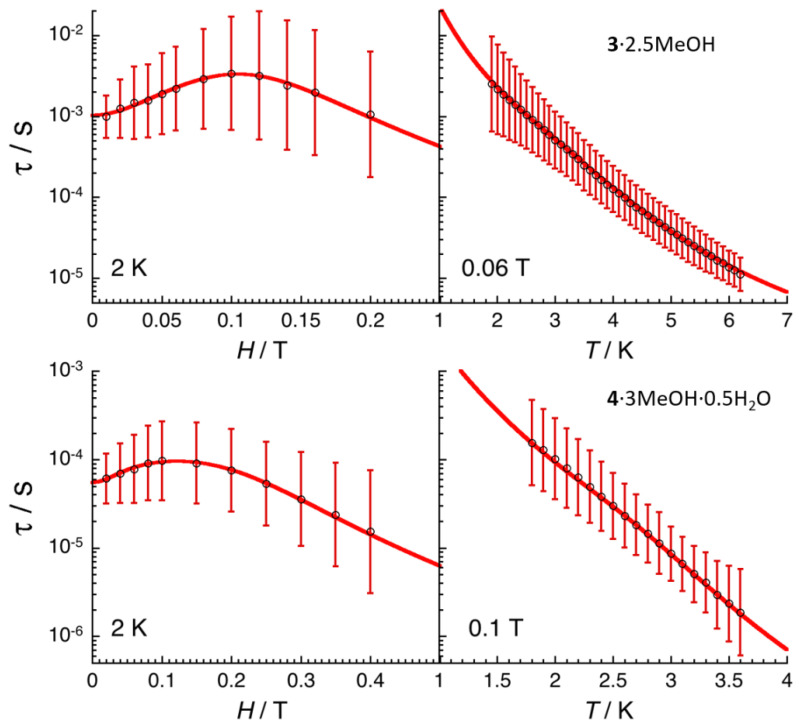
Field (left) and temperature (right) dependencies of the relaxation time (*τ*) at 2.0 K and in the presence of an applied static field of 0.06 and 1 T for complexes **3**∙2.5MeOH (top) and **4**∙3MeOH∙0.5H_2_O (bottom), respectively. The relaxation time was estimated from the generalized Debye fits of the ac susceptibility data shown in Figure 8 and Figure 9, Appendix A. The estimated standard deviations of the relaxation time (vertical solid bars) have been calculated from the α parameters of the generalized Debye fit (Appendix A) and the log-normal distribution as described in ref. [81]. The solid red lines are the best fit discussed in the text.

**Table 1 molecules-25-03153-t001:** Crystallographic data and structural refinement parameters for complexes **1**∙2MeOH∙2H_2_O, **2**∙2MeOH∙1.5H_2_O, **3**∙2.5MeOH and **4**∙3MeOH∙0.5H_2_O.

Parameter	1∙2MeOH∙2H_2_O	2∙2MeOH∙1.5H_2_O	3∙2.5MeOH	4∙3MeOH∙0.5H_2_O
Formula	C_26_H_24_Gd_2_N_12_O_15_∙2(CH_4_O)∙2(H_2_O)	(C_26_H_24_Tb_2_N_12_O_15_)_2_∙4(CH_4_O)∙3(H_2_O)	(C_27_H_26_Dy_2_N_12_O_15_)_2_∙5(CH_4_O)	(C_26_H_24_Er_2_N_12_O_15_)_2_∙6(CH_4_O)∙(H_2_O)
Formula weight	1159.19	2307.04	1151.67	2368.45
Crystal color	yellow	yellow	yellow	yellow
Crystal size, mm	0.12 × 0.11 × 0.04	0.20 × 0.17 × 0.06	0.22 × 0.15 × 0.05	0.20 × 0.17 × 0.06
Crystal system	triclinic	triclinic	triclinic	triclinic
Space group	*P*ī	*P*ī	*P*ī	*P*ī
Temperature, K	120	120	120	120
Radiation, Å	Mo Kα, 0.71073	Mo Kα, 0.71073	Mo Kα, 0.71073	Mo Kα, 0.71073
a, Å	10.4504(10)	10.3899(16)	10.3194(8)	10.3551(8)
b, Å	12.3976(12)	12.4185(18)	12.1909(9)	12.4468(10)
c, Å	17.4313(17)	17.609(3)	17.8985(14)	17.7769(13)
α, °	74.879(5)	73.749(7)	71.376(4)	71.992(4)
β, °	85.147(5)	84.738(7)	84.075(4)	83.975(4)
γ, °	68.821(4)	69.114(7)	71.252(4)	69.411(4)
Volume, Å^3^	2032.8(3)	2037.8(5)	2020.6(3)	2039.8(3)
*Z*	2	1	1	1
Calculated density, g·cm^−3^	1.894	1.880	1.913	1.928
Absorption coefficient, mm^−1^	3.325	3.531	3.758	4.176
θ_min_–θ_max_, °	1.918–29.713	2.098–28.438	2.084–25.505	2.101–30.489
Reflections collected/unique	47128/10876	30434/9998	120082/7400	44288/12236
Completeness to 2θ	0.940	0.974	0.984	0.996
*R* _int_	0.0374	0.0692	0.0600	0.0296
Refined parameters/restraints	563/3	578/2	530/1	587/2
*R*_1_[*I* > 2σ(*I*)] ^a^, *wR*_2_ ^b^ (all data)	0.0330, 0.0646	0.0545, 0.1100	0.0338, 0.0810	0.0241, 0.0554
Goodness-of-fit on *F*^2^	1.059	1.069	1.107	1.036

^a^*R*_1_ = Σ(|*F*_o_| − |*F*_c_|)/Σ|*F*_o_|. ^b^
*wR*_2_ = {Σ[w(*F*_o_^2^ − *F*_c_^2^)^2^]/Σ[w(*F*_o_^2^)^2^]}^1/2^.

**Table 2 molecules-25-03153-t002:** Selected interatomic distances (Å) and the Ln-O-Ln bond angles (°) in complexes **1**∙2MeOH∙2H_2_O, **2**∙2MeOH∙1.5H_2_O, **3**∙2.5MeOH and **4**∙3MeOH∙0.5H_2_O.

Interatomic Distances (Å)				
	Ln = Gd (**1**∙2MeOH∙2H_2_O)	Ln = Tb (**2**∙2MeOH∙1.5H_2_O)	Ln = Dy (**3**∙2.5MeOH)	Ln = Er (**4**∙3MeOH∙0.5H_2_O)
Ln1∙∙∙Ln2	4.000(1)	3.969(1)	3.945(1)	3.933(1)
Ln1-O3	2.583(3)	2.574(5)	2.586(4)	2.557(2)
Ln1-O4	2.500(3)	2.475(5)	2.468(4)	2.448(2)
Ln1-O6	2.492(2)	2.489(5)	2.468(4)	2.461(2)
Ln1-O8	2.543(3)	2.511(5)	2.485(4)	2.489(2)
Ln1-O1	2.394(2)	2.378(4)	2.359(4)	2.343(2)
Ln1-O2	2.389(2)	2.380(5)	2.364(3)	2.347(2)
Ln1-N1	2.653(3)	2.632(7)	2.614(5)	2.610(2)
Ln1-N2	2.573(3)	2.551(6)	2.523(4)	2.517(2)
Ln1-N5	2.550(3)	2.533(6)	2.534(5)	2.514(2)
Ln1-N6	2.547(3)	2.534(6)	2.512(4)	2.506(2)
Ln2-O9	2.488(3)	2.470(5)	2.462(4)	2.451(2)
Ln2-O11	2.458(3)	2.465(5)	2.436(4)	2.443(2)
Ln2-O12	2.472(3)	2.457(5)	2.443(4)	2.435(2)
Ln2-O14	2.472(3)	2.452(5)	2.439(4)	2.418(2)
Ln2-O1	2.353(2)	2.337(5)	2.326(3)	2.310(2)
Ln2-O2	2.383(2)	2.361(4)	2.348(4)	2.344(2)
Ln2-N4	2.562(3)	2.545(5)	2.509(5)	2.505(2)
Ln2-N8	2.528(3)	2.511(6)	2.503(5)	2.478(2)
Ln2-O15/O1M	2.402(3)	2.383(5)	2.383(4)	2.347(2)
C6-N2	1.289(5)	1.301(9)	1.297(7)	1.295(3)
N2-N3	1.411(4)	1.406(7)	1.403(6)	1.411(3)
N3-C8	1.294(5)	1.277(9)	1.302(7)	1.302(3)
C8-O1	1.308(4)	1.309(8)	1.299(6)	1.307(3)
C19-N6	1.290(4)	1.292(9)	1.314(7)	1.292(3)
N6-N7	1.398(4)	1.386(8)	1.396(6)	1.402(3)
N7-C21	1.303(4)	1.311(8)	1.292(7)	1.305(3)
C21-O2	1.313(4)	1.304(8)	1.309(6)	1.310(3)
**Ln-O-Ln Bond Angles (°)**				
Ln1-O1-Ln2	114.8(1)	114.7(2)	114.7(1)	115.4(1)
Ln1-O2-Ln2	113.9(1)	113.7(2)	113.7(1)	113.9(1)

**Table 3 molecules-25-03153-t003:** To date crystallographically characterized metal complexes of HL and L¯, and relevant structural information.

Compound ^a^	Coordination Mode ^b,c^	Nuclearity /Dimensionality	Coordination Geometry ^d^	Ref.
[CdBr_2_(HL)]	1.10011	Mononuclear	tbp	[49]
[Ln(NO_3_)_3_(HL)(MeOH)_2_] (Ln = La, Ce)	1.10011	Mononuclear	cpa	[50]
[Nd(NO_3_)_3_(HL)(H_2_O)]	1.10011	Mononuclear	bsa	[50]
[PdCl_2_(HL)]	1.01100	Mononuclear	sp	[51]
[HgX_2_(HL)] (X = Cl, Br)	1.10011	Mononuclear	spy	[52]
[HgI_2_(HL)(H_2_O)]	1.10011	Mononuclear	oct	[52]
{[Pb_3_Br_6_(HL)_2_]}_n_	1.10011	1D (metal-organic ribbon)	7-coordinate ^e^, oct	[53]
[PdCl(L)	1.01011	Mononuclear	sp	[51]
[PdCl(L)	1.01100	Mononuclear	sp	[51]
[Cu_4_(L)_4_(H_2_O)_2_](NO_3_)_4_	2.21011, 2.11111	Rectangular [2 × 2] grid	spy, oct	[47]
[Mn_4_(CF_3_SO_3_)(L)_4_(H_2_O)_3_](CF_3_SO_3_)_3_	2.21011	Square [2 × 2] grid	oct	[54]
[Mn_5_(L)_6_](ClO_4_)_4_	2.21011	Trigonal bipyramidal topology	oct	[54]
[Mn_4_(N_3_)_4_(L)_4_]	2.21011	Square [2 × 2] grid	oct	[55]
[Cu_4_Br_2_(L)_4_]Br_2_	2.21011, 2.11111	Rectangular [2 × 2] grid	spy, oct	[48]
[Ni_4_(NO_3_)_2_(L)_4_(H_2_O)](NO_3_)_2_	2.21011	Square [2 × 2] grid	oct	[48]
[Co_2_(L)_3_](ClO_4_)_3_	2.01111	Dinuclear helicate	oct	[48]
[Co(L)_2_](ClO_4_)	1.10011	Mononuclear	oct	[48]
[Ln_2_(ΝO_3_)_4_(L)_2_(H_2_O)] (Ln = Gd, Τb, Er)	2.21011	Dinuclear	sph, scsa	this work
[Dy_2_(NO_3_)_4_(L)_2_(MeOH)]	2.21011	Dinuclear	sph, scsa	this work

^a^ Lattice solvent molecules have been omitted. ^b^ Using the Harris notation. ^c^ The coordination modes are shown in Figure 2. ^d^ Abbreviations: sp = square planar; spy = square pyramidal; tbp = trigonal bipyramidal; oct = octahedral; cpa = capped pentagonal antiprismatic; bsa = bicapped square antiprismatic; sph = sphenocorona; scsa = spherical capped square antiprismatic.

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
