# Peer review of "Asymmetric Dinuclear Lanthanide(III) Complexes from the Use of a Ligand Derived from 2-Acetylpyridine and Picolinoylhydrazide: Synthetic, Structural and Magnetic Studies†"

_molecules, 2020, doi:10.3390/molecules25143153_

Round 1

Reviewer 1 Report

This authors presents four new asymmetric molecular compounds with potential application as molecular magnets. The introduction, the bibliography comparison and the overall characterisation are well performed.

However, there are some points that must be improved:

  • Regarding the manuscript itself:
  1. The abstract is too long: the first ten lines are more an introduction than an abstract.
  2. In page 5, equations 1 and 2 would be better described as reaction schemes.
  3. A few typos and mistakes are present in the manuscript.
  4. In the representation of the coordination polyhedra the real ones should be represented in the image (superimposed with the donor atoms), and the idealised ones, moved to the side for comparison.
  5. The representation style and the orientation of the molecules in figures S1 and S2 should be analogous.
  6. The tone of the text in some parts of the manuscript is too casual.
  • Regarding the results presented:
  1. Powder X-ray diffraction should be performed on the solids to ensure the purity of the crystalline samples. SCXRD is very useful as a characterisation technique, but the structural results are only true for the analysed crystal, not necessarily for the bulk solid.
  2. Likewise, to claim that a compound is isostructural with another (pages 4 and 5), PXRD data must be compared satisfactorily for the compounds.
  3. The crystal packing of the four structures is almost identical, despite the theoretical difference in volume and supramolecular interactions due to the supposedly coordinated MeOH molecule in the Gd derivative. Furthermore, in the structure of 3, the ellipsoid for the C atom attached to O15 is much larger that expected. This could be due to a partial occupation of this position, that would mean in practice the random cocrystallisation of both molecules with water and MeOH coordinated. However, the underlying question is that this structure cannot be compared with the other three, because it was collected at a higher temperature (which affects the structural parameters and the disorder of the molecules in the interstices), but mostly because it has been subjected to a removal of the unwanted electron density in the experimental data, in order to fit the desired model. I strongly disagree with the use of SQUEEZE, as it refines the experimental data against the model which is dubious from an ethical point of view. Most likely, a data collection at low temperature would yield an electron density map analogous to those of compounds 1, 2 and 4 and would not require the use of SQUEEZE. Consequently, the data for compound 3 should be collected again, applying the same conditions as 1, 2 and 4 in order to be able to properly compare them and to claim that the molecular species in 3 is different. 
  4. There is an important problem with the residual density near the Tb atom in the structure of 2. This could be due to the fact that the crystal was not a single domain but a twin or an aggregate. A data treatment taking into account the possibility of a second domain should be tried.

Author Response

”1. The abstract is too long: the first ten lines are more an introduction than an abstract.”

The comment is correct. We have removed the first ten lines from the previous Abstract.

“2. In page 5, equations 1 and 2 would be better described as reaction schemes.”

Since (i) The structural formula of the neutral ligand is shown in Figure 1, (ii) the coordination mode of the deprotonated ligand is incorporated in Figure 2, and (iii) the molecular structures of the complexes are illustrated in Figures 4, 5, S1 and S2, we prefer to show the formation of the complexes using chemical equations.

“3. A few typos and mistakes are present in the manuscript.”

We read carefully the ms and corrected few typos and mistakes, e.g. please see our answers to Reviewers 2 and 3 who suggested some mistakes.

“4. In the representation of the coordination polyhedra the real ones should be represented in the image (superimposed with the donor atoms), and the idealised ones, moved to the side for comparison.”

It is not obvious to us what the Reviewer means. We have drawn the coordination polyhedra with the same manner as in ~100 papers from our groups. We are asking your and Reviewer’s indulgence to retain the coordination polyhedra in their present form.

“5. The representation style and the orientation of the molecules in figures S1 and S2 should be analogous.”

The comment is correct. We changed Figure S2 which now has the same representation style and orientation with Figure S1.

“6. The tone of the text in some parts of the manuscript is too casual.”

English are not our mother language. We have improved the tone of the text in few parts.

“1. Powder X-ray diffraction should be performed on the solids to ensure the purity of the crystalline samples. SCXRD is very useful as a characterisation technique, but the structural results are only true for the analysed crystal, not necessarily for the bulk solid.”

The comment is scientifically correct. The pXRD instrument in our University has been out of order since last February. In our case, we present other evidences that the samples are pure (microanalyses, IR spectra, the value of the χT product at room temperature, the fact that the bulk solid was generated crushing single crystals and subsequent drying of the resulting powder, …).

“2. Likewise, to claim that a compound is isostructural with another (pages 4 and 5), PXRD data must be compared satisfactorily for the compounds.”

Please, see our answer above. To satisfy the Reviewer we had added the term “most probably” in the sentence “Unit-cell determination … isomorphous to 1∙2MeOH∙2H2O.”

“3. The crystal packing of the four structures is almost identical, despite the theoretical difference in volume and supramolecular interactions due to the supposedly coordinated MeOH molecule in the Gd derivative. Furthermore, in the structure of 3, the ellipsoid for the C atom attached to O15 is much larger that expected. This could be due to a partial occupation of this position, that would mean in practice the random cocrystallisation of both molecules with water and MeOH coordinated. However, the underlying question is that this structure cannot be compared with the other three, because it was collected at a higher temperature (which affects the structural paramet”ers and the disorder of the molecules in the interstices), but mostly because it has been subjected to a removal of the unwanted electron density in the experimental data, in order to fit the desired model. I strongly disagree with the use of SQUEEZE, as it refines the experimental data against the model which is dubious from an ethical point of view. Most likely, a data collection at low temperature would yield an electron density map analogous to those of compounds 1, 2 and 4 and would not require the use of SQUEEZE. Consequently, the data for compound 3 should be collected again, applying the same conditions as 1, 2 and 4 in order to be able to properly compare them and to claim that the molecular species in 3 is different.”

To satisfy this understandable comment, data for a new fresh crystal of the Dy(III) complex have been collected at 120 K. The same large disorder for most of the lattice solvent molecules has been found. A large number of constraints and restraints had to be used to give a decent model and to avoid using the SQUEEZE procedure of PLATON. The text has been changed accordingly, and details about the refinement and disorder of the solvent molecules is explained in the experimental section of the revised ms. and in the_refine_special_details of the new cif file. However, due to the large number of constraints and restraints that were needed, the exact nature and number of those lattice solvent molecules should be taken with care. We have also deleted the all reference 88 which was devoided to PLATON SQUEEZE. Please note that the formula of the Dy(III) complex has been changed, but the mass is almost the same, so that there is no need to change the magnetism part.

“4. There is an important problem with the residual density near the Tb atom in the structure of 2. This could be due to the fact that the crystal was not a single domain but a twin or an aggregate. A data treatment taking into account the possibility of a second domain should be tried.”

No twin law was detected. Examination of the reconstructed precession images do not indicate any sing of twinning or the presence of another domain. However, the formation of a small amount of ice during the collection may have affected the accuracy of the intensity of some reflections. One can also note that the largest residual electronic densities are located nearby Tb ions and cannot be due to misassigned or missing atoms. If due to another domain, this second domain is very minor.

We are grateful to Reviewer 1 for her/his valuable comments which improve the quality of the ms.

Reviewer 2 Report

This paper reports on a series of dimeric Ln(III) coordination compounds synthesized by reaction of Ln(NO3)2.nH2O and the multidentate ligand N’-(1-(pyridin-2-yl)ethylidene)pyridine-2-carbohydrazide. This work is a contribution to the field of rare earth coordination chemistry, with important achievements in the understanding of the peculiar chemical and physical properties of Ln(III) complexes. The present work is a special focus on magnetic properties with the aim to stabilize asymmetrical complexes to produce magnetic relaxation processes.

Actually, the paper constitutes a piece of work involving detailed chemistry and structural analysis of gadolinium(III), terbium(III), dysprosium(III) and erbium(III) complexes. Their magnetic properties are thoroughly investigated and the magnetic relaxation observed is convincingly discussed based on the structural features and compared to literature.

This paper deserves publication with few minor revisions:

Line 170-180: the comment on the synthesis could be shorten. For instance, without clear justification for adding a base to the reaction and/or discussing why yield improvement was expected, the comment on equation 1 & 2 does not bring much here. The comment about ref 50 (line 187-188) is enough for the discussion.

Line 262: deptononation should read deprotonation

Line 311: the plot is magnetization vs HT-1 but the measurements are M vs T.

Line 562: please let the reader taking a message home or not; Simply main or important results of this work ….. would be nice.

In SI

Line 87-88 figure S10 caption: the solid lines seem not drawn in the plots.

Author Response

“Line 170-180: the comment on the synthesis could be shorten. For instance, without clear justification for adding a base to the reaction and/or discussing why yield improvement was expected, the comment on equation 1 & 2 does not bring much here. The comment about ref 50 (line 187-188) is enough for the discussion.”

We have shortened the comment on the synthesis, as suggested.

“Line 262: deptononation should read deprotonation.”

This mistake has been corrected.

“Line 311: the plot is magnetization vs HT-1 but the measurements are M vs T.”

The text is correct. Both M vs H (at four low temperatures) and M vs HT-1 plots are presented in Figures S6-S9.

“Line 562: please let the reader taking a message home or not; Simply main or important results of this work ….. would be nice.”

The comment is correct. We have modified the relevant sentence, as suggested.

“Line 87-88 figure S10 caption: the solid lines seem not drawn in the plots.”

The comment is absolutely correct. Indeed, we forgot to replace this figure by the final one with the corresponding fits. The “Supplementary Materials” section has been updated and the correct, new Figure S10 appears.

We thank Reviewer 2 for her/his time to study the ms and the valuable comments!

Reviewer 3 Report

The manuscript "Asymmetric Dinuclear Lanthanide(III) Complexes from the Use of a Flexible Ditopic Ligand Derived  from 2-Acetylpyridine and Picolinoylhydrazide: Synthetic, Structural and Magnetic Studies" by S.P. Perlepes and co-workers presents  investigations on four dinuclear Ln(III) complexes of general formula  [Ln2(NO3)4(L)2(S)] (1-4) where Ln = Gd, Tb, Dy or Er, S =H2O or MeOH (compound 3) and L =   N’-(1-(pyridin-2-yl)ethylidene)pyridine-2-carbohydrazide, crystallized as solvates, 1∙2MeOH∙2H2O, 2∙2MeOH∙1.5H2O, 3∙MeOH∙2H2O and 4∙3MeOH∙0.5H2O, respectively. All crystalline copounds were structurally characterized by single-crystal X-ray diffraction and their magnetic properties are evaluated in details. A particular attention has been paid on a search for single molecular magnets (SMMs) properties, finally demonstrated for the dysprosium(III) and erbium(III) compounds under external dc fields of 600 and 1000 Oe, respectively. Additionally, critical comparison with structures of other compounds based on the same ligand in its neutral or anionic form has been provided and coordination polyhedra around Ln1(III) and Ln2(III) metal centers of dinuclear coordination units were evaluated with the use of the SHAPE program.

The manuscript is well constructed, clear and well written. The conclusions are well supported by the results. Additionally, the work contains satisfactory supplementary information. In general, I have no serious objections with respect to this work. Below, are just a few comments and suggestion of some minor corrections to be introduced.

1. I do not understand why IR and UV/VIS analysis was performed on solvent-free, not for crystalline samples (or for both types of samples) (p.5, line 175).  Why TGA experiment was not performed  in order to control the drying process ? This question arose in the context of the information given in p. 10, line 506.

  1. See line 167, p. 4, did you really mean Eu(III) ?
  2. It seems that HL ligand abbreviation is more appropriate than LH,
  3. aqua not aquo ligand (see for example p. 2, line 243) is more appropriate,
  4. it seems that narrow (angles values) range would be more appropriate than the short .... range (line 250, p.2),
  5. Finally, I think that manuscript title is too long. Information about flexibility and ditopic ligand character does not necessarily should be included in the title.

In conclusion, I find this manuscript interesting and have no doubt that it presents high quality research. In my opinion this work should be published after minor revision.

Author Response

1. I do not understand why IR and UV/VIS analysis was performed on solvent-free, not for crystalline samples (or for both types of samples) (p.5, line 175).  Why TGA experiment was not performed  in order to control the drying process ? This question arose in the context of the information given in p. 10, line 506.”

The comments are correct. Fortunately, we had most data and have incorporated the relevant information in Parts 2.1 and 3.3 of the revised ms.

“2. See line 167, p. 4, did you really mean Eu(III) ?”

Yes, we mean Eu(III). It seems that the Eu(III) complex is isomorphous with its Gd(III) analogue, i.e. complex 1∙2MeOH∙2H2O. This is not unexpected, since Eu and Gd are at neighboring positions in the Periodic Table of the Elements.

“3. It seems that HL ligand abbreviation is more appropriate than LH,”

We have followed the Reviewer’s suggestion and changed the ligand’s abbreviation from LH to HL throughout the whole revised ms (including figures and tables).

“4. aqua not aquo ligand (see for example p. 2, line 243) is more appropriate,”

We have corrected this mistake.

“5. it seems that narrow (angles values) range would be more appropriate than the short .... range (line 250, p.2),”

We have done the correction.

“6. Finally, I think that manuscript title is too long. Information about flexibility and ditopic ligand character does not necessarily should be included in the title.”

We agree. The comment is correct. We have shortened the title.

We thank Reviewer 3 for her/his time to study the ms and the valuable comments!

Round 2

Reviewer 1 Report

I want to thank the authors for their answers: most of the issues have been satisfactorily solved or explained.

However, PXRD is needed for the correct characterisation of the solid phases, as other techniques such as microanalyses or IR spectra are not useful in case of polymorphism (a situation that is quite frequent in coordination compounds from lanthanide metals). I understand that the situation in the past months has made things more complicated, but perhaps the samples can be sent to other facilities for analysis.